# Does Sex Matter to the Biomedical Approach in Clinical Practice Guidelines (CPGs)?: A Systematic Review of Methodology Documents Used in the Spanish National Health System

**DOI:** 10.3390/healthcare12010074

**Published:** 2023-12-28

**Authors:** Ana M. González Ramos, Gema Serrano-Gemes

**Affiliations:** 1Institute for Advanced Social Studies (IESA), The Spanish National Research Council (CSIC), 14004 Córdoba, Spain; 2Department of Nursing and Physiotherapy, Ponferrada University Campus, University of León, 24401 Ponferrada, Spain; gserg@unileon.es

**Keywords:** sex, clinical practice guidelines, evidence-based medicine, methodology, gender blind

## Abstract

Sex and gender are important variables in health, although their incorporation in medicine has been very slow. If research is sensitive and yields fruitful sex and gender evidence, these results should be included in the guidelines for clinical practices. However, literature claims that clinical practice guidelines devote very little space to these categories. The present systematic review addresses the relevance of sex and gender dimensions through methodology documents for the development of clinical practice guidelines based on three sources: the AGREE Reporting Checklist, the GRADE Handbook, and the Spanish GuíaSalud NHS Clinical Guideline Program. Findings suggest that neglecting sex and gender issues in the biomedical approach may lead to continuing to ignore relevant evidence on biological and social dimensions that do indeed influence people’s health and diseases.

## 1. Introduction

Recent research demands more attention to gender in medicine and, indeed, it has collected solid evidence on sex and gender differences in health and illness [1,2,3]. However, there is a lack of information about the extent to which sex and gender knowledge is reaching practitioners, patients, and health managers. Clinical practice guidelines (CPGs) and guideline reports are primary instruments for reaching them, guiding medical decision making, the training of new professionals, establishing authority in health care, and advising patients [4].

Although there are several articles that implore for sex and gender issues to be incorporated into CPGs [5,6], and an increasing number of research findings contribute to completing information on how diseases and healthcare may affect men and women differently, the reasons for this gap remain largely unknown. The purpose of this work is to cast light on the absence of sex and gender evidence in medical practices based on the lack of information in CPGs and the methodological approach to develop them. Our hypothesis sustains that the limited method of the biomedical model has led to the mainstreaming of the methodological approach that frames the lack of consideration of sex and gender in the formulation of clinical guidelines [7].

With this purpose, we analyze three framework documents describing clinical instrument methodologies: the AGREE Reporting Checklist [8], the GRADE Handbook for assessing the quality of evidence and the strength of recommendations based on literature reviews [9], and the Spanish GuíaSalud NHS Clinical Guideline Program [10]. The three instruments are used in the Spanish National Health System; GuíaSalud sets out the methodology for preparing CPGs, which is based on the GRADE methodology and complemented with the AGREE II Instrument [10]. The analysis addresses the biomedical approach of these handbooks in terms of the inclusion of sex, gender, and other related words to understand relevant variations on illnesses [11,12,13].

### 1.1. Clinical Evidence in Its Context

Clinical evidence has grown exponentially in the amount of scientific data available for medicine. This is a double-edged sword since it can make it quite difficult to locate the most relevant or correct information and to understand the evidence [14]. The purpose of CPGs is to summarize the available evidence through a careful methodology to assist clinical decision-making [14], assessing security and confidence guidelines for medical practice.

The Institute of Medicine (IOM) defines CPGs as follows: ‘Clinical Practice Guidelines are statements that include recommendations intended to optimize patient care. They are informed by a systematic review of evidence and an assessment of the benefits and harms of alternative care options’ [15], pp. 25–26. The IOM also points out which characteristics CPGs should contain in order to be trustworthy: (1) they should be developed by a panel of experts with a multidisciplinary character, and representatives from the main affected groups; (2) based on systematic reviews of current evidence; (3) their development must be based on a transparent and explicit process (that minimizes biases, misrepresentations, and conflicts of interest); (4) they should consider patient preferences as well as important patient subgroups; (5) they should provide clear explanations, as well as ratings of both the quality of evidence and strength of recommendations; and (6) they should be revised as necessary/appropriate. Thus, the IOM does exhibit an interest in people as individuals who make decisions, but it seems to ignore the importance of establishing the diverse impacts that diseases and treatments may have on diverse subgroups of the population.

The usefulness of CPGs lies in the fact that they have the potential not only to help synthesize available evidence and assist professionals in making the best decisions, but that they also attempt to reduce discrepancies in clinical practice and to balance quality and cost tensions in health care [16]. If well-designed CPGs are considered the pinnacle of evidence-based medicine, they should also be an unbiased standard of healthcare; according to the literature, this does not seem to be the case for many of the CPGs currently available to practitioners [16]. On the contrary, flaws can be found, including serious problems with updating information, failures in the development of these guidelines, and, consequently, in final publications. Some of the most significant flaws will be highlighted below.

First, the quality of the CPGs. According to the literature, the quality of CPGs has improved over the years, and this is clearly seen in all the domains of the AGREE II tool. However, this progress is far from complete, and there is still room for improvement in aspects such as the applicability of the CPGs and their editorial independence [17].

Second, the way in which the guidelines are created and their ongoing development are not homogenous. Throughout the world, there are many procedures, tools, and methodologies used by countries and medical associations. In this regard, as Panteli et al. have reported, no recent study has comprehensively compared how different European countries formulate and use CPGs [18]. The study conducted by Legido-Quigley et al. in 2011 developed a questionnaire to assess the desirable characteristics that should be included when developing clinical guidelines [19]. The questionnaire was launched across 29 European countries through a key informants’ network. The results showed that there was great variability in the development of clinical guidelines in terms of methodology, the agency in charge of formulation, and the final publication, which is related to the diverse grades of quality, transparency, and sophistication.

Third, CPGs, as the gold standard of evidence-based medicine, could be expected to address all the dimensions included in the current definition of health. It is worth remembering that the WHO (World Health Organization) defines health as a state of complete physical, mental, and social well-being, rejecting the idea that health is merely a lack of illness [20]. However, final CPG documents seem to focus on the biomedical model, leaving aside the social sphere, which involves the well-being of every patient according to their diverse characteristics and their associated risk factors [12]. As Maaløe et al. pointed out in their analysis of the literature on CPGs, the interest of searching for the best synthesis of medical evidence seems to have focused mainly on evidence from experimental studies [21]. Although this may seem positive because randomized trials in controlled environments are considered the highest level of research designs in biomedicine, some problems also arise, since this kind of design is not always able to answer/address all types of research questions relevant in practicing medicine [22]. For example, Bourgault states that experimental research design lacks responses to many of the questions that arise in the nursing field [22].

Fourth, McMillan Boyles et al. have pointed out that, despite the fact that CPGs should strive to eliminate health inequities through standardized approaches, sometimes they fail. Depending on the accessibility and availability of recommended resources and services, recommendations on CPGs may not only be useless in alleviating inequalities, but worsen them, for example, among people with intellectual disabilities or low socioeconomic status [23].

Finally, another noteworthy aspect is the scarce attention that CPGs devote to sex and gender variables. Thus, the systematic review on Canadian CPGs relating to non-communicable health conditions that integrate evidence on sex/gender variables showed troubling results [6]. It found that the sex-related information displayed in CPGs was quite variable in terms of the degree of specificity provided. In addition, the review evidenced that when CPGs have included contents related to sex and gender, the words are not always used appropriately. Therefore, these authors stress the need for guidelines aimed at developing CPGs to be reviewed so that they properly include the assessment and study of differences by sex and gender in medicine [6].

In this regard, the systematic literature review conducted by Gogovor et al. [24] aimed to evaluate the integration of sex and gender concepts in guidelines for writing various health research documents (including CPGs). The study examined 407 reporting guidelines listed on the EQUATOR website, where they found almost 58% of words mentioned at least once were related to sex and gender. However, only one of the guidelines analysed made fully adequate use of these terms (and it did not focus on CPGs). Moreover, it is important to note how the vast majority of the documents analyzed in this study only mentioned the terms sex or gender for the list of demographic or baseline information. This study is interesting because it analyzes the guidelines contained in EQUATOR [24]. The EQUATOR Network is an ‘umbrella’ organization that aims to improve the way in which health research studies are reported, helping to make research useful and reproducible, which includes, among others, the AGREE Reporting Checklist (one of the reporting guidelines for developing CPGs) [25].

The literature analysis shows a crucial problem since evidence-based medicine, as well as life and health science disciplines, should guarantee the knowledge and the best available evidence in order to provide the best care to patients. The lack of information about sex and gender creates a serious limitation in these studies, since they are risk factors and determine variations in patients’ illness conditions [11]. Sex and gender must be considered in every phase of the research process [26], as well as when compiling the best knowledge on medical care.

### 1.2. Development of Specific Instruments

AGREE is a tool for evaluating the quality of practice guidelines. The original instrument was published in 2003 and consisted of 23 items divided into 6 dimensions. Later, the refined AGREE II tool was published in 2010 and, based on its structure and content, the AGREE Reporting Checklist was eventually created in 2016 [27]. According to its aims, the necessary steps are set out to achieve a high-quality practice guideline, where the checklist can be used both for drafting the practice guidelines and, later, when the CPG has been completed, checking its quality [8].

The GRADE system provides the scientific community with a comprehensive, transparent, and well-articulated methodology to summarize and rank the quality of evidence supporting the recommendations given. Guideline developers use multiple and varied systems to classify the quality of evidence supporting their recommendations [28]. The GRADE system has several advantages over previous systems, including that it [29]: (a) was developed by an international group of guideline developers; (b) provides clear interpretations of strong versus weak recommendations; (c) provides clear and comprehensive criteria for grading the quality of evidence; (d) assesses the importance of the results of alternative management strategies; (e) has a well-defined separation between the quality of evidence and the strength of recommendations; (f) uses a transparent process to move from evidence to recommendations; (g) takes into account both preferences and values; (h) is useful not only for clinical guidelines, but also for systematic reviews and health technology assessments. The use of this system is widespread and is employed by more than 25 relevant organizations, including the World Health Organization, the American College of Physicians, the American Thoracic Society, and the Cochrane Collaboration [29].

In the early 21st century, in Spain, several political decisions were made that updated how evidence-based medicine would be registered in CPGs, looking for the highest level of quality and a common methodology. The GuíaSalud Program was developed with this aim, coordinated by the homonymous institution [10]. GuíaSalud is a National Health System body created in 2002, whose members include the 17 autonomous communities and the Ministry of Health [30]. The GuíaSalud Program has the following objectives: to develop CPGs using a homogeneous methodology and to contribute to methodology for the creation of these and other tools based on quality scientific evidence [30].

The common methodological framework for the formulation of clinical practice guidelines was first proposed in 2007 and then updated in 2016. It was complemented by the checklist of the Guidelines International Network (GIN) and McMaster University, as well as the AGREE II tool [10]. Regarding quality assessment and grading of the strength of recommendations, they decided to drop the SIGN methodology (Scottish Intercollegiate Guidelines Network) and adopted the newer GRADE methodology (Grading of Recommendations, Assessment, Development and Evaluation) [10].

The steps taken in Spain indicate an interest in using a homogeneous methodology from international state of the art CPGs (although there is still great heterogeneity across countries and organizations [18,19]). The GuíaSalud methodology explains the decisions it made on the new methods adopted and which were eliminated in accordance with the main international agreements. As a result, it may reproduce mainstream practices on sex and gender categories. However, fieldwork will check similarities or deviations from international guidelines.

Finally, due to everything mentioned above, this review focuses on addressing whether the methodology used in Spain to help formulate and write CPGs adequately reflects the dimensions of sex and gender.

## 2. Materials and Methods

### 2.1. Design

A review design of the Spanish methodological context for the preparation of CPGs was conducted. This design was chosen because it allows us to address highly current issues (such as the influence of sex/gender), as well as, at the same time, to offer new perspectives on the matter [31].

### 2.2. Eligibility Criteria and Information Sources

Our eligibility criteria, adjusted to our research objective, were clear and concise:-Address the main documents used for the development and writing of CPGs in the Spanish context.-Exclude those that are not currently used.-Discard documents without reference on this topic.

Taking this into account, we reviewed the documents mentioned by the programme for the development of Clinical Practice Guidelines used in the Spanish National Health System. In this way, we obtained three documents to analyze: the AGREE Reporting Checklist, the GRADE Handbook, and the GuíaSalud NHS Clinical Guideline Program. However, the SIGN methodology was discarded because it is not currently used. The case of the Guidelines International Network (GIN) was finally discarded because there is not information on the influence of sex/gender on human health (there is no mention of ‘sex’, and ‘gender’ is used only one time to suggest balance in working team).

### 2.3. Synthesis Methods

This study is based on analyzing the content of three guidelines on the production of knowledge for clinical practice: the AGREE Reporting Checklist, the GRADE Handbook, and the GuíaSalud NHS Clinical Guideline Program [8,9,10]. We conducted a quantitative collection of keywords to address the presence or lack of reference to sex and gender categories.

On the categories included, Moerman et al., in a similar work locating sex-specific evidence on clinical questions in MEDLINE, used a list of words previously created by Montgomery and Sherif [32,33]. This seminal work aimed to identify abstracts with relevant information on sex and gender differences. They found 19 terms, which we find only partially helpful because their work focused on women’s health, while the goal of this work attempts to illuminate the gap on sex-gender differences. Herein, we seek evidence that explicitly refers to men, women, and other gender categories, considering their biological and their social dimensions. As such, although we first examined these words, it required a final decision that is explained below.

After a comprehensive examination of the AGREE Reporting Checklist and the GRADE Handbook, we eventually identified two different sets of related words: direct reference words (sex, gender, men, women, male, and female) and indirect words (related to sexual orientation, pregnancy, and reproductive organs). These words were examined to check if they are present or absent in documents. In the examination of the GRADE Handbook, the word ‘setting’ appears as a keyword of interest, which was added to the collection of indirectly related words because, for this document, it highlights the sex and other social categories of patients.

Language differences entail a challenge in this analysis because there is limited gender marking in English, while many words mark gender in Spanish. This is the primary reason why there seems to be more sex and gender related words in the methodology document of GuíaSalud than in English-language documents, so we delved into this issue [10]. We looked for similar words in Spanish, finding synonymous and gender marking words (for example, the Spanish language for ‘men’ uses ‘hombre’ and ‘varón’, both related to human beings, and ‘macho’ and ‘hembra’ for plants and animals) that we included in the analysis. We discuss the relevance of the use of inclusive language and gender marking in both languages in the Section 4.

Once the keywords were identified, we moved on to analysis. We collected useful data from the keyword list, including the frequency (How many times does it appear?), contextualization through handbook sections (Where is it placed?), literality (What is the exact quotation?), and making sense of the word (What does this quote mean for wo/men? What does it try to explain about them? Is this evidence used properly or misused?).

## 3. Results

### 3.1. AGREE Reporting Checklist

The AGREE Reporting Checklist makes no mention of the words gender, woman, man, women, men, male, or female. As for the word ‘sex’, it is only mentioned in item 3 (belonging to domain 1): ‘3. Population. Describe the population (i.e., patients, public, etc.) to whom the guideline is meant to apply.’, where one of the information criteria is: ‘Target population, sex and age’ [8] (p. 1 Appendix). As for indirect words, there is no mention of sexual orientation, pregnancy, or reproductive organs, or other terms of a similar nature. However, throughout the tool, the need to indicate the type of population is mentioned on multiple occasions. Domain 4 of the checklist is one of the most interesting sections, since it mentions the need to indicate the type of population (general public or patients) when giving recommendations or for managing the different options. Certainly, in order to avoid ambiguities, the type of population must be indicated as specifically as possible, and aspects related to sex and gender must also be included. However, the checklist makes no mention of these terms beyond what is indicated above.

### 3.2. GRADE Handbook

The analysis of the GRADE Handbook evidences the lack of the social dimension in learning about health and illnesses, and a poor understanding of the contribution of the inclusion of sex and gender in biomedicine [9]. It appears a few times and there is neither a clear statement about the relevance of sex and gender categories, nor other relevant variables such as age, race, socioeconomic status, or health vulnerability. Table 1 summarizes the frequency, context, and section where they appear.

The majority of the words were used in the context of display examples (i.e., ‘One study found no statistically significant association between either funding mechanism, investigator rank, or sex and publication’ [9] (lines 1296–1297), ‘Gender and type of drug (aspirin, paracetamol, etc.) are examples of categorical variables’ (2443–2444). ‘Men’ was cited twice, plus once in singular (‘A systematic review of the effects of testosterone on erection satisfaction in men with low testosterone identified four RCT’ (621–622), showing an unclear result or introduction of risk of bias to the analysis and, on the contrary, an example of clear association that contributes to quality: ‘High quality evidence for moderate benefits of testosterone treatment in men with symptomatic androgen deficiency to improve bone mineral density and muscle strength’ (51–57 col. Examples, Table 6.3.).

‘Women’ was used in a context of examples related to maternity: ‘Hypertension in women planning conception and in pregnancy’ (35–36 col. Examples, Table 6.3.), and ‘Our recommendations reflect a belief that most women will place a low value on avoiding the pain, cost, and inconvenience of heparin therapy to avoid the small risk of even a minor abnormality in their child associated with warfarin prophylaxis’ (1997–1999). The third example of the use of the word ‘women’ was related to setting and enables the inclusion of this indirectly related word (‘Even the setting, however, can be defined as part of the definition of the population (e.g., women in low-income countries or men with myocardial infarction in a primary or rural health care setting) (282–285)’, exceptionally, the social context is cited). By contrast, ‘woman’ or ‘female’ never appears in the handbook.

In line with the previous description, other words are related to maternity or female reproduction: ‘Pregnant women’s strong aversion to even a small risk of important fetal abnormalities may be one such situation’ (1719–1720), and the female organ replaces the mention of women: ‘In managing patients with a diagnosis of cervical intraepithelial neoplasia, a precursor of prevent cervical cancer, based on visual inspection with acetic acid (VIA) clinicians may proceed to treatment directly or apply a strategy of testing for human papilloma virus and VIA’ (2157–2159).

‘Setting’ was used 35 times in the handbook with polysemic senses; setting is used as a context for tests, as countries with different health risks, and services (hospital, primary care, etc.); however, setting replaces the mention of the social dimension or population characteristic affecting the results of the diagnosis, treatment, and risk factors in Section 2, devoted to ‘Framing the health care question’ (250). The aforementioned quotation states: ‘Even the setting, however, can be defined as part of the definition of the population (e.g., women in low-income countries or men with myocardial infarction in a primary or rural health care setting)’ (282–284).

Sexual orientation also refers to examples related to men (‘the use of condoms in homosexual male relationships as a way of preventing the spread of HIV’ (1425–1426).

The goal of GRADE, enounced in the Overview, elucidates the epistemological framework: ‘A systematic search is performed to identify all relevant studies and data from the individual included studies is used to generate an estimate of the effect for each patient-important outcome as well as a measure of the uncertainty associated with that estimate (typically a confidence interval)’ (119–122). This definition highlights the absence of the sex and gender perspective because of the pre-eminence of the individual approach, linked to reducing uncertainty. The definition identifies the studies and its aim is to decide how estimation is more confident for ‘each patient’.

Individual centred analysis explains the omission of the social dimension of health. In fact, sex and gender, as well as other characteristics of the population (race, age, income, rural or urban), are placed in setting. The GRADE Handbook [9] explains that ‘A guideline question often involves another specification: the setting in which the guideline will be implemented’ (279–281), where the setting considers the features of the population.

### 3.3. GuíaSalud

The methodology document of GuíaSalud follows the GRADE methodology and individual paradigm from a biomedical model. For example, it says: ‘The term patients is used generically in the Manual and, in addition to patients, it refers to relatives, carers, users, patient representatives, patient organizations, patient federations or coalitions, mutual aid groups involving a collective, volunteers and NGOs providing care to people who suffer a health problem and citizens who may be affected by the CPG recommendations’ [10], p. 10. However, GuíaSalud uses the plural, patients, instead of the singular noun, patient, without a clear intentionality of displaying the social differences between patients involved in the studies and distinguishing the diverse impacts on men and women.

Table 2 below shows that the GuíaSalud keywords list is larger than GRADE in both the number of keywords and their frequency of appearance [10]. The number of keywords is somewhat related to gender markers in the Spanish language but, taken together, this quantitative result shows more attention to sex variables in GuíaSalud than in the GRADE methodology handbook.

Throughout the pages of the document, the keyword ‘sex’ is mentioned more times and with a clearer focus than in the GRADE Handbook. For example, Section 2 states that sex, age, pathology, and clinical status should be part of the target population (p. 15); sex, age, family context, labor setting, education, health habits, and health services are influential factors in the pathology process (p. 17); it insists on the inclusion of social dimensions (age, sex and setting) and disease factors (phase, comorbidity, risk…) as influential factors affecting the risk factors of patients (p. 39); the clinical question should include the type of patient according to sex, age, comorbidities, etc., as this is fundamental in the search for certainty information (p. 60); ‘[p]rognosis reflects the probability of an outcome occurring in people with a particular disease or condition or with a particular characteristic (age, sex or genetic profile)’. (p. 98); and, occasionally, social dimensions cause inequalities (p. 134).

The Spanish handbook states historical factors affecting the inclusion of this variable: ‘It was not until 1999 that the FDA did not mandate the need to include women and men in clinical trials in order for the trial to be considered a quality trial’ (p. 97). Likewise, it displays interest in the gender equality balance of the CPG development team (p. 25).

Regarding the section where these words appear, they were located across the handbook. It is worth explaining that the different chapters were written by diverse authors, which corroborates the relevance of saying something for the guide development team, even if they enunciate the lack of evidence (p. 97) or it states: ‘Although it is desirable to speak in terms of men and women, in this document we use generic masculine to designate all individuals without distinction of sex’ (p. 11), although this rule is not always applied. The contexts in which all the keywords appear are diverse: 43.5% were used in examples, 25.8% in conceptualization, 17.7% were repetitions in titles and references, and 12.9% due to gender inclusive language markers.

The word ‘gender’ is almost always used as a synonym of ‘sex’, showing a poor understanding of the difference. However, the handbook does distinguish female on one occasion that employs the strict meaning of the social role of women doing informal care work for the community.

One of its recommendations to optimize search evidence is: ‘Caution with searching by aspects such as age or gender. These are sensitive fields that may be affected by inaccurate indexing that may bias results’ (p. 62). Similar to the GRADE Handbook, it indicates possible unclear results or risk of bias by inclusion of the sex/gender approach in searches.

Women’s issues are important in terms of reproductive health, if we consider the number of times pregnancy or pregnancies or pregnant women are cited in the handbook (Table 2). In context, ‘women’ is used as part of a general statement (p. 97), as a relevant factor in the formulation of the PICO (Patients, Intervention, Comparison, Outcome) (Table 4.1. Components of PICO clinical questions that guide GRADE methodology) (p. 39), to cite a typical health problem, such as the importance of bisphosphonate therapy in osteoporotic women (p. 45) and mainly with reference to pregnancy or reproductive health.

Quotations with men and male (both words ‘varón’ and ‘masculino’) appear secondarily, are also linked to sexual health (p. 92), and advice on limitations of the study used in CPGs. For example, ‘The existence of serious limitations of the studies, that should be considered a risk of bias, because of low quality of evidence (e.g., bias of including only males in the study) […] except for justified cause, in case of a pathological or care process involving only males’ (p. 97).

The significant number of mentions of sex-related words seems to indicate that the Spanish Public Health Agency pays more attention to this issue. First, inclusive language is a reason to think about the importance of sex and gender. This is a result of more overall attention, with examples from the handbook and good practices from universities and other institutions in Spain; widespread awareness in Spanish society makes explicit reference to this issue. Second, there is a number of times that sex and gender are referred to as a source of noise, but while GRADE mentioned it in this sense, GuíaSalud paves the way to formulating precise PICO questions that may improve the findings declared in the final publication. This will be crucial in the next analysis proving that those CPGs where the PICO asks about sex and gender issues guide the response for these terms adequately.

### 3.4. Overview of the Three Handbooks

Table 3 displays a summary of the quantitative results found in the analysis of these three methodology guidelines (Table 3), which will be discussed below.

## 4. Discussion

Addressing the lack of sex and gender information in the methodology that guides the formulation of confident clinical guidelines is crucial for clinical practice. Many studies on medical research yield information revealing that sex and gender are indeed relevant in the illness process, preventive medicine, and health status, although this evidence is useless if it is not included in the CPG, which is why literature has demanded more attention on sex and gender inclusion [5,6,24]. This work supports the idea that more emphasis on the relevance of sex and gender in health and diseases is important to create more secure and confident recommendations on clinical evidence.

If these documents ignore the evidence on sex and gender, and intersectional social dimensions involving people’s features, evidence will misguide recommendations for clinical practice. Misinterpretation of treatment doses and different symptoms for men and women and ethnic groups or socioeconomic groups may endanger the robustness of CPGs and guideline reports [2,6,24,26]. Despite the scarce number of times that these concepts are mentioned in the three methodology documents, as revealed in Table 3, we also pay attention to the context in which they were used. We found the use of ‘setting’ limited in the AGREE and GRADE Handbook as the only reference to sex and gender dimensions with an influence on human health; evidence on research and practice medicine provides knowledge on sex, gender, and other social dimensions that have diverse effects and may be reflected in diverse recommendations [21,22].

As we pointed out in the Section 3, a challenging difference between English and Spanish guidelines is associated with the inclusive gender markers in the Spanish language. While common use of the Spanish language is quite committed to inclusive language (for example, ‘doctor’ and ‘doctora’), the working team states that it has not followed this policy and uses the generic masculine form to designate all genders indistinctly [10], p. 11. Spanish language may provide more opportunities to mention indirect forms to include sex and gender differences in health, but figures also suggest a high interest in reflecting these differences: it mentions the word ‘sex’ 10 times for different reasons and there are twice as many mentions of women than men in examples (see Table 3).

The Spanish handbook proves the importance of PICO questions to explain how findings show different results for men and women, since the definition of PICO includes some awareness on sex or gender issues, and the findings reflect these differences and gender issues. This is clear through the higher number of times GuíaSalud includes ‘sex’ or ‘gender’ compared to the GRADE and AGREE frameworks.

We wondered if the alignment of the Spanish GuíaSalud Program with the supposedly more common international methodology could create sex and gender biases, where the response is ambiguous. On the one hand, there are clear differences between languages and making a political decision on gender awareness (even if it is not well-applied) in Spain [10], p. 62. On the other, the biomedical approach, based on patients as neutral objects of research, is rooted in the framework of the three documents. As Tannenbaum et al. claim, guidelines framework may contain higher recommendations for working teams to take into account the association between sex and gender outcomes, adding more quality evidence [6].

Even if the GRADE Handbook uses patient in singular and GuíaSalud uses patients, both ignore the diverse situations deriving from sex and gender differences, as well as the social dimensions that may play a role in alleviating inequalities [23]. In general, the CPGs usually explain how comorbidity and risk factors may influence diseases but omit how sex and gender dimensions can influence diseases.

From the AGREE, GRADE, and GuíaSalud methodologies, we also need to discuss the necessity of incorporating sex- and gender-related words in every issue: how they influence or affect prevalence, treatment, prevention, and access to diagnoses and resources. Just like gender guidelines in medical research demand the incorporation of sex and gender categories in every phase of the investigation process [6,24,34,35,36], clinical guidelines should incorporate information on sex and gender, giving evidence about whether they are relevant or not in prevalence, treatment and results.

## 5. Conclusions

The analysis shows a sex- and gender-blind approach in CPG and guideline report frameworks. Although there is wide agreement on the influence of sex and gender on health and disease, it reveals the scarce attention given to including these categories for professional care when they formulate guidelines. However, the individual approach in the biomedical model makes it necessary to mention whether the patients are men or women, and whether the disease is caused by male sexual orientation, which is the case of the GRADE Handbook and GuíaSalud. However, poor utilization of sex and gender categories is clear in mainstream methodological guidelines.

This finding is relevant for a future work because it creates the framework for professionals preparing CPGs to guide clinical practices, teach new professionals, and suggest public health policies. This work also tests the keywords to identify sex and gender terms in CPGs for future studies, where an automatic tool based on scrapping analyzes sex and gender inclusion through a CPG database in the GuíaSalud Program. Scrapping is a process of importing data from websites into files or spreadsheets to extract data from the content. This method ensures that words can be counted and localised in the text for automated contextualisation, minimising the risk of mistakes or biases produced by human and manual analysis.

## Figures and Tables

**Table 1 healthcare-12-00074-t001:** Overview of keywords listed in the GRADE Handbook.

		Frequency	Context	Section
Direct	Sex	1	Example	5.2.5 Publication bias
	Gender	2	Examples	9. Glossary 2.3 Other considerations (framing the health care question)
	Men/Man	3	Examples	5.2.1 Study limitations (Risk of Bias) Table 6.3. Quality evidence 2. Framing the health care question
	Women	3	Examples	Table 6.3. Quality evidence 6.4.3 Providing transparent statements 2. Framing the health care question
	Male	2	Examples	5.3.3. Effect of plausible confounding Table 6.3. Quality evidence
Indirect	Setting	1	Definition statement	2. Framing the health care question
	Sexual orientation	1	Example	5.3.3. Effect of plausible confounding
	Pregnant	1	Example	6.3.3 Confidence in preferences
	Female organ	1	Example	7. GRADE diagnostic tests

Source: own elaboration. Absence of plural or singular forms and other terms means they do not appear.

**Table 2 healthcare-12-00074-t002:** Overview of keywords listing in GuíaSalud handbook.

Direct		Indirect	
Sex	10	Mother	3
Gender	4	Pregnancy/ pregnancies	17
Men	2	Women pregnant	4
Male (’varón’, ’masculino’)	5	Sexual orientation	1
Women	13	Sexual intercourse	2
Female (‘femenino’)	1		

Source: Own elaboration. Absence of plural or singular forms and other terms means they do not appear.

**Table 3 healthcare-12-00074-t003:** Summary of findings based on quantitative analysis of the keywords.

	Sex	Gender	Female Woman Women	Male Man Men	Indirect Mentions
AGREE	3	0	0	0	0
GRADE	1	2	3	5	4
GuíaSalud	10	4	14	7	27

Source: Own elaboration.

## Data Availability

The data presented in this study are available on request from the corresponding author.

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
