# Peer review of "Does Sex Matter to the Biomedical Approach in Clinical Practice Guidelines (CPGs)?: A Systematic Review of Methodology Documents Used in the Spanish National Health System"

_healthcare, 2023, doi:10.3390/healthcare12010074_

Round 1
Reviewer 1 Report
Comments and Suggestions for Authors
The purpose of this manuscript was to cast light on the absence of sex and gender evidence in medical practices based on the lack of information in clinical practical guidelines (CPGs) and the methodological approach to develop them.
The authors found that the CPGs based on the 3 sources did not devote much attention to sex and gender variables.
The manuscript is well written, and easy to follow. The comparison of the 3 methodological frameworks seems text-heavy. Maybe the authors can make it easy on the reader by highlighting the differences in a table.
Comments
1. Throughout the manuscript, the authors did not distinguish between sex and gender.
Author Response
Dear Reviewers,
Thank you very much for your comments and suggestions. We have reviewed our work based on the comments to improve the manuscript. In addition, we would like to add a few words in response to your comments.
We have included a table with differences on the three methodologies as Reviewer 1 suggested (p 9), but we are afraid that we cannot distinguish sex and gender uses beyond what we have already done, because the methodology papers do not do so.
Following the suggestion of Reviewers 2 and 3, we describe the methodology more clearly so that the study could be replicated, as well as explaining the type of analysis employed. We have also resolved the confusion in line 196 (Appendix) and line 363 (PICO). We would like to answer your question on the literature used in the analysis: we have used the Moerman et al. checklist, which used Montgomery and Sherif as the baseline for their analysis.
According to the comments of Reviewer 3, we have modified the subtitle as suggested and the subtitle of section 2. As we have mentioned, we expanded the Materials and Methods section in line with the changes proposed. The most important thing to highlight here is the explanation on choosing the three documents, mentioning the inclusion and exclusion criteria, and adding details on the diverse methodologies and checklists (pp 4-5). As suggested by Reviewers 1 and 3, we have included a table and have expanded the discussion section (p 9).
Please let us know if you have any other queries and/or suggestions that you may consider helpful for our article.
Best wishes,
The authors
COMMENTS REVIEWERS
REVIEWER 1
Comments and Suggestions for Authors
The purpose of this manuscript was to cast light on the absence of sex and gender evidence in medical practices based on the lack of information in clinical practical guidelines (CPGs) and the methodological approach to develop them.
The authors found that the CPGs based on the 3 sources did not devote much attention to sex and gender variables.
The manuscript is well written, and easy to follow. The comparison of the 3 methodological frameworks seems text-heavy. Maybe the authors can make it easy on the reader by highlighting the differences in a table.
Comments
- Throughout the manuscript, the authors did not distinguish between sex and gender.
REVIEWER 2
Comments and Suggestions for Authors
The article “Does Sex Matter to the Biomedical Approach in Clinical Practice Guidelines (CPGs)?” addresses a relevant topic, and it is well written and clear. Therefore, I consider that this article should be accepted for publication. However, minor revision will have to be made, especially in the “Materials and Methods”.
- Indeed, this section must be clearer. As it is, it will be very difficult to replicate the study.
- In line 196 you talk about “Appendix. Page 1”, however, I didn't find any Appendix anywhere!
- In this section, you say “This study is based on analysing the content of three guidelines…” but it is not said what type of analysis this is (e.g., is it Bardin's content analysis, or another? Please clarify).
- Do you follow Montgomery and Sherif, or not?
- I suggest that you place all the information regarding the data analysis methodology in the “Materials and Methods” section, so that it is possible to replicate the study.
- Results. Pay attention to the meaning of the acronyms mentioned. The first time these are mentioned, the acronyms must be mentioned in full. For example, what does “PICO” mean (line 363)?
REVIEWER 3
Comments and Suggestions for Authors
Thank you for this important piece of work. It is generally well written, however there are some issues.
Firstly, it is not clear from the title and subsections that this is a systematic review of three specific documents. The reporting has also not followed standard format for reporting systematic reviews. There are some recommendations for improving the methods, results and discussion section, provided as comments in the attached PDF file.
Please check the specific comments in the attached PDF file:
- Please mention the type of study in title: “A systematic review of methodology documents used in Spanish National Health System”
- Theorical Framework:
2.1 This does not look like a theoretical framework. A theoretical framework provides a particular perspective to examine a topic through psychological, social, organizational or economic theories and provides a framework to the study.
2.2 It is advised that the manuscript be structured as a standard systematic review
- Material and Methods:
3.1 This para is not suited in this section. Justification of the study should be towards the end of introduction section.
3.2 Please give justification for analysing these three documents, on what basis did you choose these three guidelines. Also mention what other guidelines there are and why they were not chosen for analysis.
- Results:
4.1 It is recommended to provide a comparison of the three guidelines in a single table to give more context to the results.
4.2 It will also be interesting to note what other studies have reported as a comparatory table, for example the reference number 6,24. This will be useful in expanding the discussion section, which at present is quite short and needs to be more robust.

Reviewer 2 Report
Comments and Suggestions for Authors
The article “Does Sex Matter to the Biomedical Approach in Clinical Practice Guidelines (CPGs)?” addresses a relevant topic, and it is well written and clear. Therefore, I consider that this article should be accepted for publication. However, minor revision will have to be made, especially in the “Materials and Methods”.
Indeed, this section must be clearer. As it is, it will be very difficult to replicate the study.
In line 196 you talk about “Appendix. Page 1”, however, I didn't find any Appendix anywhere!
In this section, you say “This study is based on analysing the content of three guidelines…” but it is not said what type of analysis this is (e.g., is it Bardin's content analysis, or another? Please clarify).
Do you follow Montgomery and Sherif, or not?
I suggest that you place all the information regarding the data analysis methodology in the “Materials and Methods” section, so that it is possible to replicate the study.
Results. Pay attention to the meaning of the acronyms mentioned. The first time these are mentioned, the acronyms must be mentioned in full. For example, what does “PICO” mean (line 363)?
Author Response

(The authors gave the same response as above.)

Reviewer 3 Report
Comments and Suggestions for Authors
Thank you for this important piece of work. It is generally well written, however there are some issues.
Firstly, it is not clear from the title and subsections that this is a systematic review of three specific documents.The reporting has also not followed standard format for reporting systematic reviews. There are some recommendations for improving the methods, results and discussion section, provided as comments in the attached PDF file.
Please check the specific comments in the attached PDF file.

Author Response

(The authors gave the same response as above.)

Round 2
Reviewer 3 Report
Comments and Suggestions for Authors
Thank you for taking the suggestions positively and incorporating in your manuscript. I have just one suggestion, Table 3 should be placed within the result section and not in the discussion section.
Author Response
Dear reviewer,
Thank you for your nice suggestion. We already have placed Table 3 within the result section.
Best wishes,